# Estimating the Minimum Sample Size for Neural Network Model Fitting—A Monte Carlo Simulation Study

**DOI:** 10.3390/bs15020211

**Published:** 2025-02-14

**Authors:** Yongtian Cheng, Konstantinos Vassilis Petrides, Johnson Li

**Affiliations:** 1Division of Psychology and Language Sciences, University College London (UCL), 26 Bedford Way, London WC1H 0AP, UK; k.petrides@ucl.ac.uk; 2Department of Psychology, University of Manitoba, Winnipeg, MB R3T 2N2, Canada; johnson.li@umanitoba.ca

**Keywords:** neural networks, sample size, ordinal dataset, predictive performance, reproducibility

## Abstract

In the era of machine learning, many psychological studies use machine learning methods. Specifically, neural networks, a set of machine learning methods that exhibit exceptional performance in various tasks, have been used on psychometric datasets for supervised model fitting. From the computer scientist’s perspective, psychometric independent variables are typically ordinal and low-dimensional—characteristics that can significantly impact model performance. To our knowledge, there is no guidance about the sample planning suggestion for this task. Therefore, we conducted a simulation study to test the performance of an NN with different sample sizes and the simulation of both linear and nonlinear relationships. We proposed the minimum sample size for the neural network model fitting with two criteria: the performance of 95% of the models is close to the theoretical maximum, and 80% of the models can outperform the linear model. The findings of this simulation study show that the performance of neural networks can be unstable with ordinal variables as independent variables, and we suggested that neural networks should not be used on ordinal independent variables with at least common nonlinear relationships in psychology. Further suggestions and research directions are also provided.

## 1. Introduction

Neural networks (NNs), which are a collection of machine learning algorithms, draw inspiration from the structure and functions of the human brain.

Neural networks with adequate width and depth—enabled by their hidden layers—can approximate any complex relationship between independent variables (IVs) and the dependent variable (DV) ([13]). This universal approximation property underpins their exceptional performance in tasks such as natural language processing ([80]) and image classification ([58]), in which there are complex features between the IVs and DV. In addition, NNs have been applied in diverse fields such as cancer prediction ([14]), transportation ([73]), engineering ([48]), and psychology ([9]).

In the field of psychology, supervised NNs are often utilized to discern patterns in psychological datasets ([71]). These NNs have not only pioneered a new direction in fully leveraging high-dimensional data but have also enhanced the performance of prediction tasks involving low-dimensional data, such as ordinal variables. Various studies by computer scientists have provided empirical evidence about the sample size requirements for high-dimensional data ([8]; [29]; [36]). However, there is a notable gap in the literature regarding empirical evidence on sample size planning for fitting NN models with low-dimensional psychometric data, particularly from a prediction stability perspective in psychology, despite its importance in study design ([51]). Therefore, this study employs a Monte Carlo simulation to provide empirical evidence on the predictive performance of supervised NNs across various sample sizes and datasets, serving as a preliminary step in sample size planning. In the following paragraphs, the term ’NNs’ will refer exclusively to supervised neural networks, with a primary focus on prediction.

This study is structured into four sections. The first section offers a general introduction to the application of NNs in psychology, emphasizing existing sample size planning suggestions from previous studies. Following this, we introduce the design of our simulation study, providing justifications and detailed descriptions for each step. The subsequent section presents the simulation’s results, along with interpretations. Finally, this paper concludes with a general discussion, highlighting this study’s contributions and limitations and suggesting directions for further research.

### 1.1. Neural Network Application in Psychological Studies

NNs in psychology are typically applied to two categories of data for prediction purposes: high-dimensional data and low-dimensional data.

First, the ability of NNs to analyze high-dimensional data, such as natural language and video, has opened new possibilities for psychological research. For instance, [78] ([78]) employed a natural generic digital footprint (i.e., Facebook likes), while [44] ([44]) used similar methods to analyze social media profile pictures. In another study, [18] ([18]) applied an NN to assess vocal stereotypes in individuals with autism. The images in [44] ([44]) and vocal patterns in [18] ([18]) were transformed into high-dimensional datasets using natural language processing ([43]) or convolutional methods ([61]). These advances in NNs and automated coding techniques are gradually replacing subjective human coding, enabling the creation of valuable high-dimensional datasets that facilitate prediction tasks in psychology.

In addition, many psychological studies utilize ordinal variables from psychometric scales (e.g., results from a 5-point Likert scale with values of 1, 2, 3, 4, or 5) as independent variables in NN models. These ordinal variables are ordered categorically and typically originate from psychometric scales with inherent measurement errors and self-correlation ([5]). Compared to datasets in various disciplines used by computer scientists, psychometric data generally exhibit lower dimensionality and higher measurement error ([34]). NNs have been leveraged in these contexts to outperform traditional regression methods ([74]). Suggested by [81] ([81]), NN models hold promise to learn and capture the behavior of highly nonlinear systems with proper accuracy and low computational efforts. These advantages cannot be achieved by common linearly structured models due to system nonlinearities and complexities.

For example, [47] ([47]) used various six-point Likert variables (from 0, indicating no risk, to 5, indicating high risk) to assess risk in child protective services, with 12,978 lines of data. They found that the NN model achieved 81% accuracy, outperforming the logistic regression model, which had 66% accuracy. [23] ([23]) employed a three-point (i.e., 0, 1, and 2) developmental behavior checklist ([20]) for predicting Autism Spectrum Disorder and found that the NN performed better (ROC = 0.93) than logistic regression (ROC = 0.88) with a balanced sample size of 638. [81] ([81]) predicted bank customer satisfaction using a 51-item, five-point scale from 436 randomly selected customers. They found that the NN had a lower mean square error (MSE) than linear models (0.44 vs. 0.6).

However, the superior performance of neural network models relies heavily on having an adequate sample size. [56] ([56]) and [53] ([53]) demonstrated that insufficient sample sizes introduce randomness, which can undermine model stability. Similarly, [56] ([56]), [36] ([36]) and [29] ([29]) have shown that inadequate sample sizes during the model fitting process lead to unstable performance. Consequently, ensuring a sufficient sample size is a critical consideration for researchers employing neural networks in their studies. This issue will be elaborated on in the next section.

### 1.2. Previous Studies in Neural Network Sample Size Planning

While numerous psychological studies already employ ordinal variables as IVs in NN model fitting, there is limited empirical evidence available to help psychologists in designing a study that aims to use NNs for model fitting. Consider a scenario in which a group of psychologists wants to create a model that uses results from psychometric scales to make predictions. They believe there are some nonlinear relationships between the IVs and DV. Yet, they do not know what exactly those relationships are. Therefore, they want to use an NN to fit this model. Then, one of the most crucial questions they need to address is as follows: how many participants do they need to recruit?

Sample size planning is a crucial aspect of psychological research design. It ensures sufficient power, controls the budget, and addresses other concerns in psychological research ([51]). Various studies in the psychology discipline have provided recommendations for sample size planning for psychometric data ([40]), focusing on the validity of different statistics, such as correlation ([65]), mediation ([27]), and CFA indexes ([46]). The empirical evidence offered by these studies has supplied important guidelines for applied researchers. Additionally, given that measurement error is common in psychometric scales ([64]), researchers have also developed numerous sample size planning techniques that take measurement error into account ([25]; [41]). For example, [6] ([6]) provided suggestions on the minimum sample size requirement for null hypothesis significance testing with the criterion of a desirable level of power based on Cronbach’s alpha level. In the following paragraph, we will review the sample size planning recommendations made by computer scientists for NNs.

Numerous studies conducted by computer scientists have addressed the sample size needed to fit various types of NN models ([3]). Some rule-of-thumb guidelines have been developed for high-dimensional data, but these suggestions are inconsistent and stem from different perspectives. For instance, [8] ([8]) proposed that the sample size needs to be at least 50 to 1000 times the number of DVs based on a criterion of 99.5% multinomial accuracy. However, [36] ([36]) suggested that the sample size should be at least 10 to 100 times the number of IVs based on the same criterion. In the meantime, these suggestions are based on findings from X-ray images or high-resolution visible images provided by NASA, which makes it doubtful whether they can applied to psychometric IVs.

Although a larger sample size can provide researchers with more confidence about the NN performance estimation reported in the study ([74]), there is no uniform sample size recommendation for fitting an NN ([63]).

These suggestions stem from different dimensions, which can result in an order of magnitude difference for a single study design. For instance, consider a scenario with three categorical IVs, each with five categories, and the researcher wants to fit a DV using an NN with three fully connected layers, each with 10 neurons. According to [8] ([8]), the sample size should be between 750 and 15,000; according to [36] ([36]), it should be between 30 and 300; and according to [29] ([29]), a sample size of 600 is needed. These suggestions are based on findings from X-ray images or high-resolution visible images provided by NASA. While these recommendations may serve as references for psychological studies analyzing high-dimensional data like natural languages ([80]) or images ([58]), studies with such high-dimensional IVs can easily achieve large sample sizes. For example, [78] ([78]) used participants’ Facebook likes to predict their personalities. With Facebook’s permission, they collected Facebook information from 86,220 participants.

Computer scientists have also conducted studies on how measurement error (i.e., noise) in the independent variables influences the sample size requirement. However, these studies mostly focus on voice and image independent variables ([45]; [68]). Some computer scientists also believe that NNs are robust to measurement error in independent variables ([82]). The only study examining the influence of measurement error on NN models in psychological application research is work conducted by [34] ([34]) from a performance perspective. In their study, they were concerned about whether the level of measurement noise (i.e., reliability of 0.3, 0.6, and 0.9) can influence the performance of the supervised machine learning model. A Monte Carlo simulation study by [34] ([34]) found that the performance (i.e., $R^{2}$) of the boosting method ([26]) cannot always outperform the linear model even if there are interaction relationships between the IVs and DV. Meanwhile, they also found that the nonlinear feature is difficult to learn for boosting models in cases with a high measurement error level in the population.

In summary, although there are some studies on the sample size requirement for NN prediction tasks, there is no applicable guideline for the minimum sample size requirement when utilizing psychometric ordinal variables as IVs to make predictions about DVs of interest to studies. Additionally, there is a growing trend for psychologists to adopt a prediction perspective when reinterpreting psychological phenomena ([19]). As a result, it is likely that more studies will be conducted using a supervised NN model that fits ordinal IVs. Consequently, this study will use a Monte Carlo simulation to test the sample size requirement for NN model fitting with Likert IVs. Two criteria will be used to determine the sample size level: ensuring the performance of the NN model is stable in replication and ensuring that the NN can outperform the linear model. Several factors are included in this study: the number of IVs, the relationship between IVs and DVs, and the coefficients in the model. The performance of the model will be evaluated based on these two criteria separately, and two minimum sample size requirements (MSSRs) that meet these criteria will be reported in the Results Section.

## 2. Design

In this section, we will discuss the simulation design of the IVs and DV at first. Then, we will discuss the procedure of grid search, which is used to select the combination of hyperparamerters of the NN. Finally, we will provide two criteria for a sufficient sample size from different perspectives, with a pilot simulation to examine whether the criteria we propose are applicable.

### 2.1. Dataset Simulation Design

At first, we will discuss the simulation of the IVs and DV. In this study, we only include Likert IVs and a continuous DV.

We use a similar design to [50] ([50]). IVs are simulated from an N(3,1) normal distribution and denoted as $x_{1},x_{2},\ldots$ in the following paragraph. After that, $x_{1},x_{2},\ldots$ are fixed by rounding up and the minimum is set at 1 and the maximum at 5; they will be denoted as $X_{1},X_{2},\ldots$ in the following paragraph. The lowercase of $x_{1},x_{2},\ldots$ are the true values behind the Likert scores, and the uppercase $X_{1},X_{2},\ldots$ are the Likert scores observed by the researchers. The error caused by the Likertized procedure (i.e., rounding up in this case) is the only error included in the IVs in this study.

Three different numbers of IV conditions are included in this study, 3, 5, and 10, to calculate the simulated DV. In all conditions, the IVs and DV have both linear and nonlinear relationships in the simulated dataset. Suppose there is only a linear relationship between the IVs and DV. The NN cannot outperform linear regression in this case. Based on the Occam’s razor principle, linear regression, as a simpler model widely used in psychology, should be chosen unless the NN model demonstrates better performance. Therefore, there is no need to apply NNs when there is only a linear relationship in the population ([74]). As a result, this study only includes the IV and DV condition that there is both a linear and nonlinear relationship between them.

Nonlinear relationships are common in psychological studies ([59]). This study includes several kinds of nonlinear relationships: two-way interaction effects ($x_{a}x_{b}$) ([49]), three-way interaction effects ($x_{a}x_{b}x_{c}$) ([15]; [70]), and quadratic effects ($x^{2}$) ([28]).

These nonlinear relationships are chosen as representative of the nonlinear relationships included in this simulation study. With these three nonlinear relationships, we aim at simulating datasets that have a complex relationship between the IVs and DV. Given their robust ability to capture these nonlinear patterns, NNs are ideally suited for this task ([2]). We expect that the NN can detect these nonlinear relationships in the model fitting easily. Specifically, we include three levels of complexity of the nonlinear relationship: simple, medium, and high.

In the simple complexity level of the nonlinear relationship, only two-way interaction effects and linear relationships are included. In the medium complexity level of the nonlinear relationship, interaction effects, quadratic effects ($x^{2}$), and linear relationships are included. In the high complexity level of the nonlinear relationship, interaction effects ($x_{a}x_{b}$), quadratic effects ($x^{2}$), three-way interaction relationships ($x_{a}x_{b}x_{c}$), and linear relationships are included.

For three complexities with three IV numbers, there are nine conditions between the IV and DV. The simulation method uses the study conducted by [42] ([42]) as a reference in the design.

The simulation formula is as follows.

Simple complexity with three IVs:(1)$$Y=cof_{1}\times x_{1}+cof_{2}\times x_{2}+cof_{3}\times x_{3}+conf_{1}\times x_{1}\times x_{2}+conf_{2}\times x_{2}\times x_{3}+coferr\times N\left(0,1\right)$$

Medium complexity with three IVs:(2)$$Y=cof_{1}\times x_{1}+cof_{2}\times x_{2}+cof_{3}\times x_{3}+conf_{1}\times x_{1}\times x_{2}+conf_{2}\times x_{2}\times x_{3}+conf_{3}\times x_{1}^{2}+conf_{4}\times x_{2}^{2}+coferr\times N\left(0,1\right)$$

High complexity with three IVs:(3)$$\begin{array}{c} Y=cof_{1}\times x_{1}+cof_{2}\times x_{2}+cof_{3}\times x_{3}+conf_{1}\times x_{1}\times x_{2}+conf_{2}\times x_{2}\times x_{3}+\\ conf_{3}\times x_{1}^{2}+conf_{4}\times x_{2}^{2}+conf_{5}\times exp\left(x_{1}\right)+coferr\times N\left(0,1\right) \end{array}$$

Simple complexity with five IVs:(4)$$Y=cof_{1}\times x_{1}+cof_{2}\times x_{2}+cof_{3}\times x_{3}+conf_{1}\times x_{1}\times x_{2}+conf_{2}\times x_{4}\times x_{5}+coferr\times N\left(0,1\right)$$

Medium complexity with five IVs:(5)$$Y=cof_{1}\times x1+cof_{2}\times x2+cof_{3}\times x3+conf_{1}\times x1\times x2+conf_{2}\times x4\times x5+conf_{3}\times x_{1}^{2}+conf_{4}\times x_{5}^{2}+coferr\times N\left(0,1\right)$$

High complexity with five IVs:(6)$$\begin{array}{c} Y=cof_{1}\times x_{1}+cof_{2}\times x_{2}+cof_{3}\times x_{3}+conf_{1}\times x_{1}\times x_{2}+conf_{1}\times x_{4}\times x_{5}+\\ conf_{2}\times x_{1}^{2}+conf_{3}\times x_{5}^{2}+conf_{4}\times exp\left(x_{4}\right)+coferr\times N\left(0,1\right) \end{array}$$

Simple complexity with ten IVs:(7)$$\begin{array}{c} Y=cof_{1}\times x_{1}+cof_{2}\times x_{2}+cof_{3}\times x_{3}+cof_{4}\times x_{4}+cof_{5}\times x_{5}+\\ \;cof_{6}\times x_{6}+conf_{1}\times x_{7}\times x_{8}+conf_{2}\times x_{9}\times x_{1}0+coferr\times N\left(0,1\right) \end{array}$$

Medium complexity with ten IVs:(8)$$\begin{array}{c} Y=cof_{1}\times x_{1}+cof_{2}\times x_{2}+cof_{3}\times x_{3}+cof_{4}\times x_{4}+cof_{5}\times x_{5}+\\ \;cof_{6}\times x_{6}+conf_{1}\times x_{7}\times x_{8}+conf_{2}\times x_{9}\times x_{1}0+conf_{3}\times x_{5}^{2}+conf_{4}\times x_{1}0^{2}+coferr\times N\left(0,1\right) \end{array}$$

High complexity with ten IVs:(9)$$\begin{array}{c} Y=cof_{1}\times x_{1}+cof_{2}\times x_{2}+cof_{3}\times x_{3}+cof_{4}\times x_{4}+cof_{5}\times x_{5}+cof_{6}\times x_{6}+\\ \;conf_{1}\times x_{7}\times x_{8}+conf_{2}\times x_{9}\times x_{10}+conf_{3}\times x_{5}^{2}+conf_{4}\times x_{10}^{2}+conf_{5}\times x_{9}+coferr\times N\left(0,1\right) \end{array}$$

Here, $x_{1}\ldots x_{10}$ are the IVs, *Y*s are DVs, $cof_{x}$ is the coefficient for linear relationships, $conf_{x}$ is the coefficient for nonlinear relationships, in which x=1,2,3…, coferr is the coefficient of the error, and N(0,1) is a random number simulated from a standard normal distribution.

Due to the time limitation in simulation, the orthogonal experimental design for item coefficients can not be applied because the time consumption associated with this design far exceeds the acceptable limits for the simulation, as this is a high-compute-intensive task, which will be explained in detail later.

Therefore, we include two conditions for these two coefficients separately: small or large. A coefficient with a small condition is simulated from a uniform distribution of [0.1, 0.3], and a coefficient with a large condition is simulated from a uniform distribution of [0.5, 1]. A gap between these two distributions is designed to explore the relationship between the MSSR and categorical coefficient conditions. All these coefficients are simulated from a selected random seed and are saved in the supplementary document. There are 2×2=4 conditions of coefficients in total, in which each coefficient is simulated independently by the conditions. This means a condition with small linear relationship coefficients and large nonlinear relationship coefficients can be calculated by(10)$$Y=0.29\times x_{1}+0.28\times x_{2}+0.13\times x_{3}+0.6_{1}\times x_{1}\times x_{2}+0.7\times x_{2}\times x_{3}+0.8_{1}\times x_{1}^{2}+0.75_{2}\times x_{2}^{2}+coferr\times N\left(0,1\right)$$

Here, $x_{1},x_{2},x_{3}$ are the continuous IVs without Likertization and Y Is the simulated DV.

For all formulas in the Appendix A, coferr in the formula is the noise coefficient and is a multiple with a number simulated by a normal distribution of N(0,1). There are three levels of noise coefficients: 1,4, and 10. This design aims to add different levels of variance that can not be explained (i.e., noise).

To sum up, there are nine different kinds of relationships between the IVs and DV, four different kinds of coefficients, and three different error levels in this study. As a result, there are 144 conditions in this simulation. After *Y* is calculated with the true value of IVs like $x_{1},x_{2},\ldots$, the DV *Y* will be combined with the corresponding IVs like $X_{1},X_{2},\ldots$ to create the dataset for NN model fitting. All these coefficients are simulated from a selected random seed and are saved in a supplementary document.

Based on the design proposed above, we can calculate several statistics for the simulation datasets. These statistics will be used in the next section to interpret the results.

The theoretical maximum explainable variance $R_{t}^{2}$ is calculated by a linear regression to *Y* with all the items. For the example in (10), because $X_{i}$ is the best estimation of $x_{i}$ available in the dataset, $R_{t}^{2}$ is calculated by the regression models with factors $X_{1}$,$X_{2}$,$X_{3}$,$X_{1}\times X_{2}$, $X_{2}\times X_{3}$, and $X_{2}^{2}$ in this sample. As all factors used to calculate *Y* are included in the model, $R_{t}^{2}$ should be the theoretical maximum explainable variance.

In addition, we also calculate the prediction performance calculated by linear regression with all items included in the model, and we denote the variance that can be explained as $R_{l}^{2}$, in which *l* stands for linear. In the sample, the $R_{l}^{2}$ is calculated with a linear regression model that includes factors $X_{1}$,$X_{2}$, and $X_{3}$, which are all the items included in this study. The performance of an ideal NN model should be expected to outperform $R_{l}^{2}$.

### 2.2. Neural Network Design

Researchers have a high degree of freedom in designing NN model fitting ([16]), particularly when it comes to selecting a combination of hyperparameters for optimal performance. Hyperparameters are parameters that significantly influence the model fitting process. However, unlike regular parameters, hyperparameters cannot be determined during model fitting; they must be decided beforehand ([19]).

In practice, researchers can choose hyperparameters through grid search ([22]) combined with cross-validation ([16]). Grid search is a technique used in machine learning for hyperparameter tuning, where all possible combinations of predefined hyperparameter values are evaluated to find the optimal settings for a model ([22]). For each condition, the best combination of hyperparameters is applied to the NN using this grid search. Below, we will provide a general introduction to the hyperparameters included in the grid search, with a focus on how they can influence the performance of the NN.

#### 2.2.1. Neural Network Shape

The shape of the NN delineates its architecture in terms of the number of layers and the number of neurons (or nodes) in each layer. It is a fundamental hyperparameter that dictates the complexity and capacity of the model. An appropriate network shape is crucial: too shallow or with few neurons might lead to underfitting, while too deep or with many neurons can lead to overfitting and increased computational demands ([79]). In the meantime, the selection of shape influences the MSSR based on the research of ([3]). A larger sample size is needed if more neurons and layers are in the NN model for a stable result ([29]).

In this study, we consider only the simplest NN: a fully connected unidirectional NN, which is the model used in [23] ([23]); [47] ([47]); [81] ([81]). Therefore, we include the conditions of (10), (10, 10), and (10, 10, 10) for the NN. For instance, (10, 10) means there are two fully connected hidden layers with 10 neurons in each layer for the NN.

Based on our simulated dataset, a fully connected unidirectional neural network is sufficient for this prediction task. Yet, it should be mentioned that for more complex applications, advanced architectures are preferable: Convolutional Neural Networks (CNNs) excel with image and spatial data ([11]), while Recurrent Neural Networks (RNNs) are ideal for handling sequential data ([37]) and language and context modeling ([52]).

#### 2.2.2. Learning Rate

The learning rate is a crucial hyperparameter in neural networks that determines the step size during the optimization process. A suitable value ensures efficient convergence during training, whereas values that are too large or too small can hinder model performance by causing overshooting or slow convergence, respectively ([79]). We include three levels of learning rate in this search, 0,0001, 0.001, and 0.01 ([57]; [72]), using the Adam gradient descent method ([38]).

#### 2.2.3. Patience

Patience determines the number of epochs the training process should wait without observing improvement in a chosen metric before halting the training ([66]). This prevents overfitting and can potentially reduce training time. We include three levels of patience in this search: 5, 10, and 15 ([7]; [24]; [31]).

#### 2.2.4. Batch Size

Batch size refers to the number of training examples utilized in one iteration. It plays a pivotal role in optimizing the training process, influencing the model’s generalization ability, training speed, and convergence ([79]). While smaller batches can provide a regularizing effect and lower generalization error, larger batches can accelerate the learning process by leveraging computational efficiencies. We include three levels of batch size in this search: 32, 64, and 128 ([35]; [54]; [75]; [77]).

An orthogonal design was used in this search, which means there were 3×3×3×3=81 combinations of hyperparameters in the search. With cross-validation, the combination of hyperparameters with the best performance on the validation dataset was selected. As a result, this is a computationally intensive task. For each combination of hyperparameters, ten NN models will be fitted, and each replication in a single simulation involves 81 hyperparameter combinations.

### 2.3. Criteria for Adequate Sample Size in Neural Network Model Fitting

In this study, we use two criteria for determining the MSSR: we want the majority of the predictive performances of the NN in 1000 replications to be close to the maximum performance, and we want the majority of the predictive performances of the NN in 1000 replications to outperform linear regression.

#### 2.3.1. The Criterion Based on the Theoretical Maximum Performance

There are existing Monte Carlo simulation studies aiming to estimate the influence of sample size on the performance of linear regression in psychological studies. Linear regression is a commonly used prediction method for continuous IVs in psychology. As supervised NNs are also used for prediction, the criteria of these existing studies can be used as a reference.

For example, [60] ([60]) proposed four criteria for a stable performance of linear regressions: (i) small optimism in predictor effect estimates as defined by a global shrinkage factor of larger than 0.9; (ii) small absolute difference of less than 0.05 in the apparent and adjusted $R^{2}$; (iii) precise estimation (a margin of error less than 10% of the true value) of the model’s residual standard deviation; and, similarly, (iv) precise estimation of the mean predicted outcome value (model intercept). Except for the second criterion, none of the other criteria can be applied to the performance of the NN.

Therefore, we propose a criterion for this study based on the design discussed in [60] ([60]), with some modifications for Monte Carlo simulation. We expanded the absolute value and doubled the tolerance range from 0.05 to 0.1. For an adequate sample size, we expect that 95% of NNs in the simulation will have a performance that will meet the criterion of(11)$$R_{97.5\%}^{2}-R_{2.5\%}^{2}<0.1$$

Here, $R_{97.5\%}^{2}$ is the 97.5% percentile of $R^{2}$ in the simulation; $R_{2.5\%}^{2}$ is the 2.5% percentile of $R^{2}$ in the simulation.

With this design, we aim to find an MSSR for the NN that ensures that most of the NN models neither overperform nor underperform in relation to $R_{t}^{2}$. As noted earlier, $R_{t}^{2}$ is computed using the model that simulated the DV. Consequently, $R_{t}^{2}$ represents the theoretical upper bound of performance. In the following paragraphs, we will refer to this criterion as the criterion based on the theoretical maximum performance.

#### 2.3.2. The Criterion Based on Outperforming of the Linear Model

We also propose another criterion: the NN model should outperform the linear regression model. We propose this metric for these two reasons:

1: Based on the Occam’s razor principle, if a complex model like an NN cannot outperform the simple linear model, the NN model should not be chosen, as the linear regression model offers a simpler interpretation. Psychological researchers also suggest that the performance of the supervised machine learning fitted model should be compared with the performance of the linear model in psychological research to make a binary decision on whether to use the supervised machine learning method in a study ([63]).

2: Given that there are both linear and nonlinear relationships between the IVs and DV in this simulation study, the NN should outperform the linear model.

Therefore, we aim to find an MSSR such that 80% of the NN models can outperform the linear model, which is calculated by including all the items linearly in a regression. The performance of the linear model is calculated with a sample size of 100,000. An adequate sample size should make 80% of $R_{l}^{2}>R_{NN}^{2}$. In the following paragraph, we will call this criterion the criterion based on outperforming the linear model.

### 2.4. General Simulation Design

The whole simulation study was conducted in Python ([55]), with the packages Tensorflow ([1]) and Keras ([10]). We included sample sizes of 1000, 2500, 5000, 10,000, and 20,000. If either of the two criteria could not be met with a sample size of 20,000, we then tested additional sample sizes of 25,000, 30,000, 35,000, 40,000, 45,000, and 50,000, one after another. If a sample size of 50,000 was still insufficient to meet either of the two criteria, we considered the sample size requirement for this criterion in this condition as unattainable.

After the dataset was simulated, 80% of the simulated data were randomly assigned to the training dataset, and the rest were assigned to the testing dataset. In total, 20% of the training dataset was randomly assigned as the validation dataset for the hyperparameters selection by grid search. All the performance reported in the study is based on the performance of the model on the testing dataset.

## 3. Results

The full results for all sample sizes are provided in the supplementary document. A selection of these results is presented in Table 1. The MSSR based on the theoretical maximum performance is identified, and the MSSR based on outperforming the linear model is provided in Table 2.

Below, we will discuss the MSSR results provided by the two criteria. We will report Spearman’s correlation coefficients between the MSSR and the rank factors included in this simulation study. Both the correlation coefficients and the *p*-values will be provided. However, we emphasize that these *p*-values should be viewed as a reference only, given the low sample size in these tests ([12]).

### 3.1. Criterion Based on the Theoretical Maximum Performance

Based on my findings, an MSSR can be determined for most of the conditions in this study, using the theoretical maximum performance as the criterion. The conditions where stable performance close to $R_{t}^{2}$ in replication could not be found are mainly those with a low $R_{t}^{2}$ (e.g., condition 25: $R_{t}^{2}=0.0273$).

For the conditions where an MSSR has been found using this criterion (N = 96), the MSSR is positively correlated with error level (r = 0.5392, *p* < 0.001) and number of IVs (r = 0.1035, *p* = 0.3181). The MSSR is negatively correlated with linear coefficients (r = −0.066, *p* = 0.5255), complexity level (r = −0.136, *p* = 0.1886), and nonlinear coefficients (r = −0.2611, *p* = 0.0106, and $R_{t}^{2}$ (r = −0.832, *p* < 0.001). Moreover, if a researcher aims to restrict the $R_{t}^{2}$ to within a margin of ±0.05% using a sample size of 1000, the neural network would need to explain 80% of the variance—a level of explanatory power that is rarely achieved in psychological research.

### 3.2. Criterion Based on the Outperforming of the Linear Model

We have found unexpected results for the MSSR with this criterion. In all the conditions simulated in this study, there are both linear and nonlinear relationships between the IVs and DV. We expected the performance of the NN to be better than the linear model, as the linear model can only fit the linear relationship between the IVs and DV. However, we found that in only 89 out of 108 conditions, we could find an MSSR of less than 50,000 to ensure that 80% of the results provided by the NN were better than those provided by linear regression. While previous research has shown that high measurement error levels in the independent variable can linearize nonlinear relationships ([34]), we did not anticipate that the measurement error inherent in the Likert process would be sufficient to cause neural network performance to mirror that of a linear model. Furthermore, this measurement error may affect hyperparameter selection, thereby further impacting the overall performance of the model ([69]).

The conditions where an MSSR could not be identified using this criterion all had an $R_{t}^{2}<0.1$. These conditions also tended to have lower complexity and smaller nonlinear coefficients.

For the conditions where an MSSR was found using this criterion (N = 90), MSSR was positively correlated with error level (r = 0.37, *p* < 0.001), number of IVs (r = 0.14, *p* = 0.5299), and linear coefficients (r = 0.11, *p* = 0.1956). MSSR was negatively correlated with complexity level (r = −0.32, *p* = 0.002), nonlinear coefficients (r = −0.35, *p* < 0,0001), and $r_{t}$ (r = −0.72, *p* < 0,0001).

In the meantime, the variance can be explained by the model (i.e., $R^{2}$), which should be inversely proportional to the width of the interval, based on both theoretical proof and empirical evidence ([21]; [30]). However, a universal constant based on this relationship cannot be found across all conditions. Therefore, we cannot provide a rule-of-thumb formula for researchers to use for NN sample size planning based on the simulation results of this study.

## 4. Discussion, Limitations, and Further Directions

This study was expected to offer guidance regarding the necessary sample size for fitting NN models with psychometric data. However, we found that meeting both criteria is challenging unless the sample size is quite large: a stable result close to the maximum theoretical performance cannot be achieved even with a sample size of 10,000 in some conditions when using the NN. In other conditions, even a stable performance cannot be reached with a sample size of 50,000. This indicates that the NN requires a very large sample size to learn some common nonlinear relationships in psychology. Therefore, based on the results of this study, we do not recommend using an NN for at least some prediction tasks with ordinal IVs.

Specifically, NNs should not be applied to datasets with a high level of inexplicable noise. In these datasets, a sample size of 50,000 is insufficient to meet the criteria proposed in this study. We advise against fitting NN models to datasets where only low performance is achievable. This recommendation is not because an NN model cannot outperform classical regression in these conditions; in fact, good performance can sometimes be observed due to sampling error. However, this instability is precisely why we advise against this application. While an NN model may dramatically outperform the linear regression model in the testing dataset due to sampling error, we cannot confidently expect a replication study with the same sample size (i.e., 1000) to yield a similar performance advantage.

By making this statement, we are critiquing the reproducibility of NNs with psychometric data unless the sample size is large enough. While psychologists consider performance estimation from independent testing datasets with cross-validation as more reliable and view it as a potential solution to the replication crisis ([39]; [62]), this study found that the results provided by the NN may not be replicable even when the training/testing dataset division is applied, unless an adequate sample size is available. However, if a necessary sample size can be recruited for a study, researchers can identify nonlinear relationships using developed methods like [33] ([33]), and a regression model with nonlinear terms can be conducted, which eliminates the necessity of using an NN.

To the best of our knowledge, researchers may still want to apply NNs to predict factors of interest in studies with psychometric IVs if they believe that the relationships between IVs and DVs are complex, cannot be captured by an analytical formula, or are far from linear relationships, such as exponential ($e^{x}$) relationships ([28]). In the meantime, NNs may still be useful to provide stable performance in the case that the reliability of IVs is high (i.e., the measurement error of IVs is at a low level). Yet, more research should be conducted in this field. In addition, CNNs or RNNs can still be useful when analyzing high-dimensional data, such as images, as regression cannot deal with this kind of high-dimensional data.

Regarding limitations and future directions, the most significant limitation of this study is the restricted range of conditions examined. Future research should encompass a broader variety of conditions for NN models and dataset designs. It is acknowledged that simulating every possible condition of NN models in a single study is unfeasible. Nevertheless, future studies should explore additional factors, such as different types of relationships between IVs and DVs and varying sample size thresholds. Moreover, future research should also consider scenarios involving binary or multinomial DVs. Unlike multinomial logistic regression, NN algorithms do not necessitate the assumption of a linear relationship between the IVs and the logit transformation of the DV. This characteristic could be a potential advantage of NN models, offering a more flexible approach to handling various data types and relationship dynamics.

However, researchers should be aware that this study utilized GPU boost with multi-threading in simulation, employing 16 threads on a high-performance server, and still required five days to complete the simulation. Therefore, similar tasks can be time-consuming, as we have mentioned above.

The discovery that NNs cannot provide stable results has sparked new considerations regarding the choice of supervised machine learning methods, particularly for low-dimensional ordinal IVs. While NNs are sophisticated and hold great potential for predicting outcomes with high-dimensional IVs, psychologists might also explore other advanced methods rooted in regression. Specifically, penalized linear regression algorithms, which balance bias and variance for improved performance, are noteworthy alternatives. Lasso ([67]), Ridge ([32]), and Elastic Net ([83]) are three penalized linear regression models known to often surpass traditional linear regression in performance. While these models have fewer hyperparameters and coefficients, they might need a smaller sample size for a stable performance near the true value.

According to the simulation results of this study, in scenarios with measurement errors and high noise levels—even in the presence of nonlinear relationships—these advanced linear regression methods can yield robust performance and are highly recommended. Another advantage of these methods is their higher interpretability compared to NN models, a factor contributing to their popularity among psychologists ([4]; [17]; [76]). For quantitative psychologists, an additional research avenue could involve gathering empirical evidence to assist applied psychologists in planning sample sizes for penalized regression methods. This approach could significantly enhance the accuracy and efficacy of research in psychology.

## Figures and Tables

**Table 1 behavsci-15-00211-t001:** A selection of full results.

Cond	1	19	26	38	98
Lp1000	0.9392	0.7884	0.4269	0.9590	0.1774
Lp2500	0.9618	0.8928	0.6477	0.9750	0.5197
Lp5000	0.9726	0.9197	0.7760	0.9815	0.6566
Lp10000	0.9813	0.9454	0.8486	0.9882	0.7479
lp20000	0.9882	0.9620	0.8885	0.9924	0.8455
Lp25000		0.9649	0.9023		0.8424
Lp30000		0.9667	0.9111		0.8538
Lp35000		0.9714	0.9090		0.8593
Lp40000		0.9706	0.9258		0.8808
Lp45000		0.9748	0.9208		0.8697
Lp50000		0.9765	0.9312		0.8761
Up1000	1.0325	1.1462	1.3220	1.0173	1.4159
Up2500	1.0197	1.0977	1.1837	1.0110	1.2486
Up5000	1.0155	1.0632	1.1413	1.0094	1.1807
Up10000	1.0124	1.0480	1.0929	1.0064	1.1430
Up20000	1.0084	1.0337	1.0710	1.0052	1.0994
Up25000		1.0276	1.0553		1.0679
Up30000		1.0272	1.0568		1.0578
Up35000		1.0248	1.0439		1.0480
Up40000		1.0221	1.0437		1.0488
Up45000		1.0221	1.0424		1.0457
Up50000		1.0213	1.0410		1.0463
Above1000	0.6260	0.2780	0.1710	0.5810	0.3970
Above2500	0.8640	0.3110	0.0880	0.8860	0.3190
Above5000	0.9720	0.4210	0.1250	0.9740	0.2350
Above10000	0.9950	0.4820	0.1830	0.9940	0.2050
Above20000	0.9980	0.5870	0.4050	0.9990	0.2250
Above25000		0.7110	0.6430		0.1660
Above30000		0.7950	0.7010		0.1460
Above35000		0.8000	0.7420		0.1500
Above40000		0.8630	0.7650		0.1400
Above45000		0.8910	0.8210		0.1670
Above50000		0.9120	0.8350		0.1540

Note: Cond refers to the condition selection as an example in this study, in which 1 stands for condition 1 in Table 2, 19 stands for condition 19 in Table 2, 26 stands for condition 26 in Table 2, 38 stands for condition 38 in Table 2, and 98 stands for condition 98 in Table 2; LPx, in which x is 1000, 2500, 5000, 10,000, 20,000, 30,000, 35,000, 40,000, 45,000, or 50,000, refers to the 2.5% percentile of the $\frac{R_{n}^{2}}{R_{t}^{2}}$; Upx, in which x is 1000, 2500, 5000, 10,000, 20,000, 30,000, 35,000, 40,000, 45,000, or 50,000, refers to the 97.5% percentile of the $\frac{R_{n}^{2}}{R_{t}^{2}}$, and the criterion based on the theoretical maximum performance is calculated by the Lpx and Upx; and Abovex, in which x is 1000, 2500, 5000, 10,000, 20,000, 30,000, 35,000, 40,000, 45,000, or 50,000, refers to the percentage of $R_{n}^{2}>R_{l}^{2}$ in 1000 replications. The missing value in sample sizes of 30,000, 35,000, 40,000, 45,000, and 50,000 means that a sample size that can satisfy both criteria can be found in a sample size of 1000, 2500, 5000, 10,000, or 20,000.

**Table 2 behavsci-15-00211-t002:** Simulation results of the minimum sample size requirement and intervals of 1000 sample size.

Cond	Complex	Linear	Nonliner	Error	IVnumber	$R_{t}^{2}$	MSSRR	MSSRA
1	1	1	1	1	3	0.7193	1000	2500
2	2	1	1	1	3	0.8335	1000	5000
3	3	1	1	1	3	0.8764	1000	1000
4	1	1	2	1	3	0.8639	1000	1000
5	2	1	2	1	3	0.9009	1000	1000
6	3	1	2	1	3	0.9042	1000	1000
7	1	2	1	1	3	0.8161	1000	5000
8	2	2	1	1	3	0.8546	1000	5000
9	3	2	1	1	3	0.8968	1000	1000
10	1	2	2	1	3	0.8941	1000	2500
11	2	2	2	1	3	0.9052	1000	2500
12	3	2	2	1	3	0.9066	1000	2500
13	1	1	1	4	3	0.2422	20,000	20,000
14	2	1	1	4	3	0.3789	10,000	20,000
15	3	1	1	4	3	0.7144	1000	1000
16	1	1	2	4	3	0.6901	1000	2500
17	2	1	2	4	3	0.7918	1000	2500
18	3	1	2	4	3	0.8792	1000	1000
19	1	2	1	4	3	0.3228	10,000	40,000
20	2	2	1	4	3	0.4658	5000	20,000
21	3	2	1	4	3	0.6600	1000	5000
22	1	2	2	4	3	0.6840	1000	5000
23	2	2	2	4	3	0.8456	1000	20,000
24	3	2	2	4	3	0.8779	1000	1000
25	1	1	1	10	3	0.0273	X	X
26	2	1	1	10	3	0.1006	X	45,000
27	3	1	1	10	3	0.3801	10,000	5000
28	1	1	2	10	3	0.2037	20,000	20,000
29	2	1	2	10	3	0.5920	2500	5000
30	3	1	2	10	3	0.7862	1000	1000
31	1	2	1	10	3	0.0571	X	X
32	2	2	1	10	3	0.1021	X	X
33	3	2	1	10	3	0.1788	20,000	35,000
34	1	2	2	10	3	0.2720	10,000	20,000
35	2	2	2	10	3	0.4595	5000	10,000
36	3	2	2	10	3	0.7874	1000	1000
37	1	1	1	1	5	0.6414	2500	10,000
38	2	1	1	1	5	0.8140	1000	2500
39	3	1	1	1	5	0.8874	1000	1000
40	1	1	2	1	5	0.8615	1000	2500
41	2	1	2	1	5	0.8997	1000	2500
42	3	1	2	1	5	0.9019	1000	2500
43	1	2	1	1	5	0.7892	1000	5000
44	2	2	1	1	5	0.8416	1000	10,000
45	3	2	1	1	5	0.8809	1000	2500
46	1	2	2	1	5	0.8772	1000	2500
47	2	2	2	1	5	0.8988	1000	2500
48	3	2	2	1	5	0.9034	1000	1000
49	1	1	1	4	5	0.0901	X	X
50	2	1	1	4	5	0.2098	20,000	30,000
51	3	1	1	4	5	0.5490	5000	2500
52	1	1	2	4	5	0.5162	5000	5000
53	2	1	2	4	5	0.7472	1000	2500
54	3	1	2	4	5	0.8655	1000	2500
55	1	2	1	4	5	0.1650	20,000	X
56	2	2	1	4	5	0.4475	2500	20,000
57	3	2	1	4	5	0.5848	2500	5000
58	1	2	2	4	5	0.5867	2500	10,000
59	2	2	2	4	5	0.7823	1000	2500
60	3	2	2	4	5	0.8758	1000	2500
61	1	1	1	10	5	0.0236	X	X
62	2	1	1	10	5	0.1009	50000	X
63	3	1	1	10	5	0.2910	10,000	10,000
64	1	1	2	10	5	0.1678	25,000	20,000
65	2	1	2	10	5	0.4803	5000	10,000
66	3	1	2	10	5	0.7546	1000	1000
67	1	2	1	10	5	0.0450	X	X
68	2	2	1	10	5	0.0875	X	X
69	3	2	1	10	5	0.2500	20,000	20,000
70	1	2	2	10	5	0.2405	20,000	20,000
71	2	2	2	10	5	0.4895	5000	10,000
72	3	2	2	10	5	0.7649	1000	2500
73	1	1	1	1	10	0.5467	2500	20,000
74	2	1	1	1	10	0.8163	1000	5000
75	3	1	1	1	10	0.8539	1000	2500
76	1	1	2	1	10	0.8735	1000	2500
77	2	1	2	1	10	0.8920	1000	2500
78	3	1	2	1	10	0.9017	1000	1000
79	1	2	1	1	10	0.7371	5000	50,000
80	2	2	1	1	10	0.8356	1000	10,000
81	3	2	1	1	10	0.8652	1000	5000
82	1	2	2	1	10	0.8596	1000	5000
83	2	2	2	1	10	0.8965	1000	5000
84	3	2	2	1	10	0.9039	1000	1000
85	1	1	1	4	10	0.0725	X	X
86	2	1	1	4	10	0.3495	10,000	25,000
87	3	1	1	4	10	0.5593	5000	5000
88	1	1	2	4	10	0.4726	5000	10,000
89	2	1	2	4	10	0.7625	1000	5000
90	3	1	2	4	10	0.8517	1000	2500
91	1	2	1	4	10	0.2199	20,000	X
92	2	2	1	4	10	0.3829	10,000	40,000
93	3	2	1	4	10	0.6038	2500	10,000
94	1	2	2	4	10	0.5092	5000	20,000
95	2	2	2	4	10	0.7743	1000	10,000
96	3	2	2	4	10	0.8759	1000	2500
97	1	1	1	10	10	0.0180	X	X
98	2	1	1	10	10	0.0509	X	X
99	3	1	1	10	10	0.1588	40,000	X
100	1	1	2	10	10	0.0970	25,000	X
101	2	1	2	10	10	0.4152	5000	20,000
102	3	1	2	10	10	0.6640	2500	2500
103	1	2	1	10	10	0.0379	X	X
104	2	2	1	10	10	0.1086	X	X
105	3	2	1	10	10	0.1574	35,000	X
106	1	2	2	10	10	0.1875	20,000	X
107	2	2	2	10	10	0.4790	5000	10,000
108	3	2	2	10	10	0.6319	2500	5000

Note: ‘Cond’ refers to the condition in the simulation; ‘Complex’ indicates the complexity of the nonlinear relationship between IVs and the DV, where ‘1’ signifies simple complexity, ‘2’ medium complexity, and ‘3’ high complexity; ‘Linear’ denotes the categorical linear coefficient level, with ‘1’ representing coefficients simulated from the range of 0.1 to 0.3, and ‘2’ from 0.5 to 0.1; ‘Nonlinear’ refers to the categorical nonlinear coefficient level,
with ‘1’ for coefficients from 0.1 to 0.3, and ‘2’ for coefficients from 0.5 to 0.1; ‘IVnumber’ refers to the number of independent variables; ‘$R_{t}^{2}$’ refers to the maximum variance that can be theoretically explained; ‘MSSRR’ refers to the minimum sample size required for the stability of the performance criterion; and ‘MSSRA’ refers to =the minimum sample size for the criterion of outperforming the linear model, with an ‘X’ indicating that the necessary sample size to meet this criterion cannot be found

## Data Availability

The original contributions presented in this study are included in the link of in https://osf.io/ptzkj/, accessed on 6 February 2025. Further inquiries can be directed to the corresponding author(s).

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
