# Peer review of "Estimating the Minimum Sample Size for Neural Network Model Fitting—A Monte Carlo Simulation Study"

_behavsci, 2025, doi:10.3390/bs15020211_

Round 1

Reviewer 1 Report

Comments and Suggestions for Authors

The paper discusses the use of neural networks (NN) in psychological studies, particularly focusing on how to determine the minimum sample size required for effective model fitting. The authors conducted a simulation study to evaluate NN performance with different sample sizes and propose criteria for optimal sample planning.

With few adjustments based on the suggestions provided, this paper has the potential to make a strong impact in the emerging field.

  • The authors are invited to present equations 1 and 2 in a proper way in the paper, other than the appendix.
  • The authors are invited to clarify the acceptable limits of the simulation on page 4.
  • The authors are invited to clarify the effect of the uniform distribution representing the data, as it can be source of errors and to clarify why “.4” is not considered.
  • The authors are claiming that NN cannot have a minimum dataset to operate and be performant with a minimum dataset of ordinal data. They are invited to justify why the simplest NN is used while the dataset used is a combination of linear and nonlinear data representation.
  • The authors are invited to clarify the representation of numbers 1, 19, 26, 38 and 98 in table 1

Author Response

Dear reviewer:

Thank you very much for your suggestions for making this paper better. Below, we will reply to your suggestions one by one. we have also attached the revised manuscript.

With few adjustments based on the suggestions provided, this paper has the potential to make a strong impact in the emerging field.

  • The authors are invited to present equations 1 and 2 in a proper way in the paper, other than the appendix.

Thank you. All of the simulation formulas has been rearranged from Equation (1) to Equation (9) on Page 10 to Page 11 in the new manuscript.

  • The authors are invited to clarify the acceptable limits of the simulation on page 4.

After carefully considering this criterion, we have changed it in this new manuscript. We now have used a criterion based on R^2 rather than r, as R^2 is more likely to be reported as a metric for model performance in psychology. The new criterion is similar to the old criterion based on r, which makes the result is consistent with the original manuscript.

We have not re-conducted the simulation study. However, we have re-selected the MSSR based on this new criterion.

We have also changed the result in Table 1 and Table 2, as well as other parts of this study, based on this new criterion. However, the MSSRs on all conditions are generally consistent, and the major conclusion of this study is not influenced by this change.

We have also provided a detailed explanation of this new criterion:

As we have mentioned on Page 16, fifth paragraph:

For example, Riley et al. (2019) have proposed four criteria for a stable performance of linear regressions: (i) small optimism in predictor effect estimates as defined by a global shrinkage factor of larger than 0.9; (ii) small absolute difference of less than 0.05 in the apparent and adjusted R2; (iii) precise estimation (a margin of error less than 10% of the true value) of the models residual standard deviation; and similarly, (iv) precise estimation of the mean predicted outcome value (model intercept). Except for the second criterion, none of the other criteria can be applied to the performance of NN.

Therefore, We propose a criterion for this study based on the design discussed in Riley et al. (2019) with some modifications for Monte Carlo simulation. We expanded the absolute value and doubled the tolerance range from 0.05 to 0.1. For an adequate sample size, we expect that 95% of NN in the simulation perform that will meet the criterion of

R2_97.5%- R2_2.5%<.1

In which R2_97.5% is 97.5% percentile of R2 in the simulation, R2_2.5% is 2.5% percentile of R2 in the simulation.

The authors are invited to clarify the effect of the uniform distribution representing the data, as it can be source of errors and to clarify why “.4” is not considered.

Thank you for your questions. We hope the categorical coefficient level could have a certain level of difference to provide a clearer result in later correlation analysis. Suppose a fully covered distribution of [.1 , .5] and [.5 , 1] are applied for these two conditions separately. One may select a value of .48, and the other can be a value of .51, which eliminates the difference between the two.

Page 12, third paragraph:

Therefore, we include two conditions for these two coefficients separately: small or

large. A coefficient with a small condition is simulated from a uniform distribution of [.1,

.3], and a coefficient with a large condition is simulated from a uniform distribution of [.5, 1]. A gap between these two distributions is designed to explore the relationship between MSSR and categorical coefficient conditions.

  • The authors are claiming that NN cannot have a minimum dataset to operate and be performant with a minimum dataset of ordinal data. They are invited to justify why the simplest NN is used while the dataset used is a combination of linear and nonlinear data representation.

Thank you for this question. We have provided two separate sections to answer this question and show that a simple NN is necessary but not overcomplex for this problem.

On page 9, fifth paragraph:

These nonlinear relationships are chosen as representative of the nonlinear relationships included in this simulation study. With these three nonlinear relationships,we aim at simulating datasets that have a complex relationship between IVs and DV. Given their robust ability to capture these nonlinear patterns, NN is ideally suited for this task. (Almeida, 2002).

Page 14, fourth paragraph:

Based on our simulated dataset, a fully connected unidirectional neural network is sufficient for this prediction task. Yet, it should be mentioned that for more complex applications, advanced architectures are preferable: Convolutional Neural Networks (CNNs) excel with image and spatial data (Chua, 1997), while Recurrent Neural Networks (RNNs) are ideal for handling sequential data (Khaldi et al., 2023), language and context modeling (Mikolov et al., 2011).

  • The authors are invited to clarify the representation of numbers 1, 19, 26, 38 and 98 in table 1

Thank you. These are the just condition numbers.

As we have mentioned in the note in Table 1:

The Cond is the selected condition as an example in this study, in which 1 stands for condition 1 in Table 2, 19 stands for condition 19 in Table 2, 26 stands for condition 26 in Table 2, 38 stands for condition 38 in Table 2 and 98 stands for condition 98 in Table 2.

Reviewer 2 Report

Comments and Suggestions for Authors

The abstract is well structured, starting with an introduction to the problem, followed by the methodology, main findings and implications. However, the writing could benefit from adjustments to improve fluency and accuracy of communication. The purpose is clearly explained: to explore the use of neural networks (NNs) in psychometric data sets and to provide guidance on the minimum sample size needed. This is well defined, although I think the novel contribution to previous studies could be emphasised more. In terms of methodology, the abstract mentions the simulation approach, which is adequate, but does not sufficiently describe the design of the simulation or the specific characteristics of the data used (e.g. the range of sample sizes tested or the types of non-linear relationships simulated). In addition, there are certain fragments that need stylistic revision for clarity. For example:

·       The psychometric independent variables are usually ordinal, with comparatively low dimensions and measurement error.’ This sentence could be rephrased to indicate how these characteristics affect the modelling.

·       The finding of this simulation study shows the performance of neural networks can be unstable...’ .Should be ‘The findings of this simulation study show...’

Regarding the introduction of the article, it is well structured and addresses the necessary elements to contextualise the problem, highlight the gap in knowledge, and justify the study. However, I must point out that some fragments contain repetition and excess of words that could be simplified to improve fluency and clarity:

For example:

·       NNs with sufficient width and depth, incorporating hidden layers, can generally fit any complex model [? ]. Consequently, they exhibit exceptional performance in various tasks, including natural language processing (NLP) [? ] and image classification [? ]’.

Although various applications of NNs are mentioned, some statements lack detail or specific references, as in:

·       NNs have been leveraged in these contexts to outperform traditional regression methods [? ].

It would be useful to cite specific examples or explain why they outperform traditional methods.

I also find that the transition between paragraphs is not always smooth. For example, the change from the general description of the NNs to the sample size planning studies is abrupt. Including transition sentences would help guide the reader. On the other hand, although previous literature on sample size requirements in NNs is cited, some examples seem out of context for psychometric studies. For example, recommendations based on high-resolution imaging data (NASA) do not align well with the focus of the study. Finally, as for the introduction, it seems excessively long, which might discourage the reader. Subsection 1.2 (Previous Studies in Neural Network Sample Size Planning) could be summarised to focus on the key ideas.

As for point 2 design, the section is well structured and covers the main aspects of simulation design and the methodology used. However, the text is dense and uses technical terms that may not be easily understood by readers with no experience in neural network simulations. It would be useful to include a summary outline or a table describing the general flow of the experimental design. The inclusion of linear and non-linear relationships in IVs and DVs is well substantiated, but references cited as ‘[?]’ are not specified, which makes it difficult to verify and follow up claims. This is a major problem in scientific articles. References should be complete and correctly linked. Furthermore, although the text describes in detail the generation of independent variables (IVs) and dependent variables (DVs), it would be useful to add an introductory paragraph explaining with a concrete example how the simulated variables relate to the psychological scenarios to be modelled. This would make the design more tangible for readers. As for the hypotheses, the statement that ‘NN should not outperform linear regression when the relationship is purely linear’ is based on the principle of parsimony (Occam's razor). However, it would be useful to cite studies that empirically demonstrate this relationship in psychological contexts. In terms of sample size the criteria defined for the minimum required sample size (MSSR) are clear and appropriate, but the difference of the confidence interval of 0.08 in the criterion based on maximum performance could be further justified, as is done for the criterion of exceeding the linear model. This would ensure that readers understand the relevance of this threshold.

On the other hand, some terms such as ‘likertized’ could be better explained, as it is a technical term that may not be widely known. Also, phrases such as ‘For elaboration convenience’ could be rewritten in a more formal style, such as ‘For ease of explanation.’ In addition, I think it would be useful to include a paragraph outlining the practical implications of the experimental design. For example, how might this help psychologists decide when to use NN versus more traditional methods? .Finally, with regard to this heading, the authors should review: Line 234-252 and justify why a fully connected NN is used and not other types such as CNN or RNN. With these revisions, the section could be significantly improved in terms of clarity, scientific rigour and usefulness to readers.

The presentation of results is well structured and divides the analyses into distinct sections: tables, specific discussions for each criterion (based on theoretical maximum performance and outperformance of the linear model). The tables are labelled appropriately, making it easy to track the results. They also include a wide range of relevant data (MSSR, linear and non-linear coefficients, complexity, etc.) and the explanation of technical terms in the footnotes (such as Lpx, Upx, and Abovex) helps to interpret the results. I should point out that although correlations are discussed, there is not enough discussion of their practical significance or implications for future studies. Furthermore, in section 3.2, the existence of unexpected results is mentioned without providing a clear hypothesis on the possible causes. This leaves the reader with unanswered questions. And the term ‘criterion based on theoretical maximum performance’ could be more explicit - does it refer to the best fit of the model or to another type of performance?

Regarding the last discussion section, the section presents a detailed and well-structured discussion of the implications, limitations and future directions related to the use of neural networks (NN) in psychometric data. However, there are areas that could be improved for clarity, rigour and cohesion. It should be more clearly specified which prediction tasks should be avoided or under which specific conditions NNs are inappropriate. This will help readers to interpret the scope of the recommendation. In addition, further analysis of the implications of computational time could be included, especially for researchers with limited resources. While the section addresses relevant points, the transition between key findings and recommendations could be smoother. For example, move from discussing specific results (such as NN's unstable performance) to suggesting alternatives (penalised regression) in a more structured way, explicitly highlighting how these alternatives address NN's limitations. While some future directions are mentioned, these are somewhat general. It would be good to propose specific studies, such as the assessment of NN with different hyperparameter configurations or in particular psychometric domains, to inspire more focused research.

Finally, the absence of the bibliographical references section limits the ability to fully assess the academic rigour of the article.

In summary, the article ‘Estimating the Minimum Sample Size for Neural Network Model Fitting - A Monte Carlo Simulation Study’ addresses a relevant and pressing issue in the application of neural networks (NN) to psychometric data, offering valuable insights into the challenges and limitations associated with determining appropriate sample sizes. Although the study is methodologically robust and provides detailed discussions of its findings, there are areas where clarity, organisation and contextualisation could be improved. For example, the summary and introduction are more concise, technical terms and methodologies are better explained, and the discussion of practical implications and future lines of research is improved. In addition, the absence of a reference section considerably limits the assessment of the scientific basis of the article. If these aspects were addressed, I believe that the article could better serve the intended audience by providing clearer guidance and inspiring more specific research in this field.

Author Response

Dear reviewer:

Thank you very much for your suggestions for making this paper better. Below, We will reply to your suggestions one by one.

The abstract is well structured, starting with an introduction to the problem, followed by the methodology, main findings and implications. However, the writing could benefit from adjustments to improve fluency and accuracy of communication. The purpose is clearly explained: to explore the use of neural networks (NNs) in psychometric data sets and to provide guidance on the minimum sample size needed. This is well defined, although I think the novel contribution to previous studies could be emphasised more. In terms of methodology, the abstract mentions the simulation approach, which is adequate, but does not sufficiently describe the design of the simulation or the specific characteristics of the data used (e.g. the range of sample sizes tested or the types of non-linear relationships simulated). In addition, there are certain fragments that need stylistic revision for clarity. For example:

  • The psychometric independent variables are usually ordinal, with comparatively low dimensions and measurement error.’ This sentence could be rephrased to indicate how these characteristics affect the modelling.
  • The finding of this simulation study shows the performance of neural networks can be unstable...’ .Should be ‘The findings of this simulation study show...’

Thank you for your suggestions. We have made several modifications in the abstract.

Therefore, we conducted a simulation study to test the performance of NN with different sample sizes and the simulation of both linear and nonlinear relationships.

From the computer scientist's perspective, psychometric independent variables are typically ordinal and low-dimensionala characteristic that can significantly impact model performance.

The findings of this simulation study show the performance of neural networks can be unstable with ordinal variables as independent variables, and we suggested that neural networks should not be used on ordinal independent variables with at least common nonlinear relationships in psychology.

Regarding the introduction of the article, it is well structured and addresses the necessary elements to contextualise the problem, highlight the gap in knowledge, and justify the study. However, I must point out that some fragments contain repetition and excess of words that could be simplified to improve fluency and clarity:

For example:

  • NNs with sufficient width and depth, incorporating hidden layers, can generally fit any complex model [? ]. Consequently, they exhibit exceptional performance in various tasks, including natural language processing (NLP) [? ] and image classification [? ]’.

Thank you very much for raising these concerns. At first, possibly due to a compilation error, it appears that none of my references were visible on your end. To prevent this issue from happening again, I will include a PDF file with this revised version.

We have made some modifications based on your suggestions to make the following paragraph more clearer.

On page 4, third paragraph:

First, the ability of NN to analyze high-dimensional data, such as natural language and video, has opened new possibilities for psychological research. For instance, Youyou et al. (2015) employed a natural generic digital footprint (i.e., Facebook Likes), while Liu et al. (2016) used similar methods to analyze social media profile pictures. In another study, Dufour et al. (2020) applied NN to assess vocal stereotypes in individuals with autism. The images in Liu et al. (2016) and vocal patterns in Dufour et al. (2020) were transformed into high-dimensional datasets using Nature Language Processing (Liddy, 2001) or convolutional methods (Romanyuk, 2016). These advances in NN and automated coding techniques are gradually replacing subjective human coding, enabling the creation of valuable high-dimensional datasets that facilitate prediction tasks in psychology.

Although various applications of NNs are mentioned, some statements lack detail or specific references, as in:

  • NNs have been leveraged in these contexts to outperform traditional regression methods [? ].

It would be useful to cite specific examples or explain why they outperform traditional methods.

Thank you for your suggestions. We have added an explanation provided by Zeinalizadeh et al. (2015).

On page 5, first paragraph:

NNs have been leveraged in these contexts to outperform traditional regression methods (Yarkoni & Westfall, 2017). Suggested by Zeinalizadeh et al. (2015), NN models hold promise to learn and capture the behavior of highly nonlinear systems with proper accuracy and low computational efforts. These advantages cannot be achieved by common linearly structured models due to system nonlinearities and complexities.

I also find that the transition between paragraphs is not always smooth. For example, the change from the general description of the NNs to the sample size planning studies is abrupt. Including transition sentences would help guide the reader.

Thank you for your suggestions. We have added a paragraph between them for better connection:

On page 5, third paragraph.

However, the superior performance of neural network models relies heavily on having an adequate sample size. Pecher et al. (2024) and Rajput et al. (2023a) have demonstrated that insufficient sample sizes introduce randomness, which can undermine model stability. Similarly, Haykin (2009), Kavzoglu and Mather (2003), and Rajput et al.(2023b) have shown that inadequate sample sizes during the model fitting process lead to unstable performance. Consequently, ensuring a sufficient sample size is a critical consideration for researchers employing neural networks in their studies. This issue will be elaborated on in the next section.

On the other hand, although previous literature on sample size requirements in NNs is cited, some examples seem out of context for psychometric studies. For example, recommendations based on high-resolution imaging data (NASA) do not align well with the focus of the study. Finally, as for the introduction, it seems excessively long, which might discourage the reader. Subsection 1.2 (Previous Studies in Neural Network Sample Size Planning) could be summarised to focus on the key ideas.

Thank you. We also believe these examples would not be interesting to psychologists, so we have deleted this paragraph and put the most important information into the above paragraph.

On page 7, third paragraph

These suggestions stem from different dimensions, which can result in an order of magnitude difference for a single study design. For instance, consider a scenario with 3 categorical IVs, each with 5 categories, and the researcher wants to fit a DV using an NN with 3 fully connected layers, each with 10 neurons. According to Cho et al. (2015), the sample size should be between 750 and 15,000; according to Kavzoglu and Mather (2003), it should be between 30 and 300; and according to Haykin (2009), a sample size of 600 is needed. These suggestions are based on findings from X-ray images or high-resolution visible images provided by NASA. While these recommendations may serve as references for psychological studies analyzing high-dimensional data like natural languages (Zalake & Naik, 2019) or images (Rawat & Wang, 2017), studies with such high-dimensional IVs can easily achieve large sample sizes. For example, Youyou et al. (2015) used participantsFacebook Likes to predict their personalities. With Facebooks permission, they collected Facebook information from 86,220 participants.

As for point 2 design, the section is well structured and covers the main aspects of simulation design and the methodology used. However, the text is dense and uses technical terms that may not be easily understood by readers with no experience in neural network simulations. It would be useful to include a summary outline or a table describing the general flow of the experimental design. The inclusion of linear and non-linear relationships in IVs and DVs is well substantiated, but references cited as ‘[?]’ are not specified, which makes it difficult to verify and follow up claims. This is a major problem in scientific articles. References should be complete and correctly linked. 

Thank you for raising this issue. We have provided a checked manuscript this time, as we mentioned above.

Furthermore, although the text describes in detail the generation of independent variables (IVs) and dependent variables (DVs), it would be useful to add an introductory paragraph explaining with a concrete example how the simulated variables relate to the psychological scenarios to be modelled.

Thank you for your suggestion. We have provided a general discussion about why the IVs and DV we simulated are based on the real relationship in psychometric datasets.

On page 9, fourth paragraph.

Nonlinear relationships are common in psychological studies (Richardson et al.,2017). This study includes several kinds of nonlinear relationships: the two-way interaction effects (xaxb) (Mathieu et al., 2012), the three-way interaction effects (xaxbxc) (Dawson & Richter, 2006; Wei et al., 2007), and the quadratic effects(x2) (Guastello, 2001).

This would make the design more tangible for readers. As for the hypotheses, the statement that ‘NN should not outperform linear regression when the relationship is purely linear’ is based on the principle of parsimony (Occam's razor). However, it would be useful to cite studies that empirically demonstrate this relationship in psychological contexts.

Thank you for your suggestions. We have put an explanation above in other psychology studies:

On page 5, first paragraph:

NNs have been leveraged in these contexts to outperform traditional regression methods (Yarkoni & Westfall, 2017). Suggested by Zeinalizadeh et al. (2015), NN models hold promise to learn and capture the behavior of highly nonlinear systems with proper accuracy and low computational efforts. These advantages cannot be achieved by common linearly structured models due to system nonlinearities and complexities.

In terms of sample size the criteria defined for the minimum required sample size (MSSR) are clear and appropriate, but the difference of the confidence interval of 0.08 in the criterion based on maximum performance could be further justified, as is done for the criterion of exceeding the linear model. This would ensure that readers understand the relevance of this threshold.

After carefully considering this criterion, we have changed it in this new manuscript. We now have used a criterion based on R^2 rather than r, as R^2 is more likely to be reported as a metric for model performance in psychology. The new criterion is similar to the old criterion based on r, which makes the result is consistent with the original manuscript.

We have not re-conducted the simulation study. However, we have re-selected the MSSR based on this new criterion.

We have also changed the result in Table 1 and Table 2, as well as other parts of this study, based on this new criterion. However, the MSSRs on all conditions are generally consistent, and the major conclusion of this study is not influenced by this change.

We have also provided a detailed explanation of this new criterion:

As we have mentioned on Page 16, fifth paragraph:

For example, Riley et al. (2019) have proposed four criteria for a stable performance of linear regressions: (i) small optimism in predictor effect estimates as defined by a global shrinkage factor of larger than 0.9; (ii) small absolute difference of less than 0.05 in the apparent and adjusted R2; (iii) precise estimation (a margin of error less than 10% of the true value) of the models residual standard deviation; and similarly, (iv) precise estimation of the mean predicted outcome value (model intercept). Except for the second criterion, none of the other criteria can be applied to the performance of NN.

Therefore, We propose a criterion for this study based on the design discussed in Riley et al. (2019) with some modifications for Monte Carlo simulation. We expanded the absolute value and doubled the tolerance range from 0.05 to 0.1. For an adequate sample size, we expect that 95% of NN in the simulation perform that will meet the criterion of

R2_97.5%- R2_2.5%<.1

In which R2_97.5% is 97.5% percentile of R2 in the simulation, R2_2.5% is 2.5% percentile of R2 in the simulation.

On the other hand, some terms such as ‘likertized’ could be better explained, as it is a technical term that may not be widely known. Also, phrases such as ‘For elaboration convenience’ could be rewritten in a more formal style, such as ‘For ease of explanation.’ In addition, I think it would be useful to include a paragraph outlining the practical implications of the experimental design. For example, how might this help psychologists decide when to use NN versus more traditional methods? 

Thank you. We have edited the phrase to a more formal style:

Page 9, second paragraph:

The error caused by the likertized procedure (i.e., round-up in this case) is the only error included in IV in this study.

The other one has been deleted.

In the meantime, the conclusion shows that the result of NN can still be unstable. Not too many application suggestions can be given. Yet, we have written more:

Page 21, second paragraph:

In the meantime, NN may still be useful to provide stable performance in the case the reliability of IVs is high (i.e., the measurement error of IVs in a low level.). Yet, more researches should be done in this field. In addition, CNN or RNN can still be useful when analyzing high-dimensional data, such as images, as regression cannot deal with this kind of high-dimensional data.

Finally, with regard to this heading, the authors should review: Line 234-252 and justify why a fully connected NN is used and not other types such as CNN or RNN. With these revisions, the section could be significantly improved in terms of clarity, scientific rigour and usefulness to readers.

We have added a detailed discussion about when CNN or RNN are applicable.

On page 14, fourth paragraph

Based on our simulated dataset, a fully connected unidirectional neural network is sufficient for this prediction task. Yet, it should be mentioned that for more complex applications, advanced architectures are preferable: Convolutional Neural Networks (CNNs) excel with image and spatial data (Chua, 1997), while Recurrent Neural Networks (RNNs) are ideal for handling sequential data (Khaldi et al., 2023), language and context modeling (Mikolov et al., 2011)

The presentation of results is well structured and divides the analyses into distinct sections: tables, specific discussions for each criterion (based on theoretical maximum performance and outperformance of the linear model). The tables are labelled appropriately, making it easy to track the results. They also include a wide range of relevant data (MSSR, linear and non-linear coefficients, complexity, etc.) and the explanation of technical terms in the footnotes (such as Lpx, Upx, and Abovex) helps to interpret the results. I should point out that although correlations are discussed, there is not enough discussion of their practical significance or implications for future studies.

Based on what we have found, I do not recommend using NN with the psychometric dataset. We have tried our best to provide some practical suggestions for psychologists.

Page 19, first paragraph

Moreover, if a researcher aims to restrict the R2t to within a margin of ˘.05% using a sample size of 1000, the neural network would need toexplain 80% of the variancea level of explanatory power that is rarely achieved in psychological research.

Furthermore, in section 3.2, the existence of unexpected results is mentioned without providing a clear hypothesis on the possible causes. This leaves the reader with unanswered questions. 

Thank you for your suggestions. We have provided some explanation in

Page 19, second paragraph:

While previous research has shown that high measurement error levels in the independent variable can linearize nonlinear relationships (Jacobucci & Grimm, 2020a), we did not anticipate that the measurement error inherent in the Likert process would be sufficient to cause neural network performance to mirror that of a linear model. Furthermore, this measurement error may affect hyperparameter selection, thereby further impacting the overall performance of the model (Tsamardinos et al., 2015).

And the term ‘criterion based on theoretical maximum performance’ could be more explicit - does it refer to the best fit of the model or to another type of performance?

Thank you, we have made this term more clearer.

Page 17, second paragraph:

With this design, we aim to find an MSSR for NN that ensures most of the NN models neither overperform nor underperform than R2t . As noted earlier, R2 t is computed using the model that simulated the DV. Consequently, R2t represents the theoretical upperbound of performance. In the following paragraphs, we will refer to this criterion as the criterion based on the theoretical maximum performance.

Regarding the last discussion section, the section presents a detailed and well-structured discussion of the implications, limitations and future directions related to the use of neural networks (NN) in psychometric data. However, there are areas that could be improved for clarity, rigour and cohesion. It should be more clearly specified which prediction tasks should be avoided or under which specific conditions NNs are inappropriate. This will help readers to interpret the scope of the recommendation. In addition, further analysis of the implications of computational time could be included, especially for researchers with limited resources. While the section addresses relevant points, the transition between key findings and recommendations could be smoother. For example, move from discussing specific results (such as NN's unstable performance) to suggesting alternatives (penalised regression) in a more structured way, explicitly highlighting how these alternatives address NN's limitations. While some future directions are mentioned, these are somewhat general. It would be good to propose specific studies, such as the assessment of NN with different hyperparameter configurations or in particular psychometric domains, to inspire more focused research.

Thank you very much for your suggestions. We have clarified the scernaio that NN should not be applied, which is in this study:

Page 20, third paragraph.

Specifically, NN should not be applied to datasets with a high level of inexplicable noise. In these datasets, a sample size of 50,000 is insufficient to meet the criteria proposed in this study. We advise against fitting NN models to datasets where only low performance is achievable. This recommendation is not because an NN model cannot outperform classical regression in these conditions; in fact, good performance can sometimes be observed due to sampling error. However, this instability is precisely why we advise against this application. While an NN model may dramatically outperform the linear regression model in the testing dataset due to sampling error, we cannot confidently expect a replication study with the same sample size (i.e., 1000) to yield a similar performance advantage

In addition, we have also revised the CNN and RNN we mentioned above for proper usage in the discussion.

Page 21, second paragraph:

To the best of our knowledge, researchers may still want to apply NN to predict factors of interest in studies with psychometric IVs if they believe that the relationships between IVs and DVs are complex, cannot be captured by an analytical formula, or are far from linear relationships, such as exponential (ex) relationships (Guastello, 2001). In the meantime, NN may still be useful to provide stable performance in the case the reliability of IVs is high (i.e., the measurement error of IVs at a low level.). Yet, more research should be done in this field. In addition, CNN or RNN can still be useful when analyzing high-dimensional data, such as images, as regression cannot deal with this kind of high-dimensional data.

We have also provided a more detailed discussion about the penalized regression we have mentioned in :

Page 22, second paragraph

The discovery that NN cannot provide stable results has sparked new considerations regarding the choice of supervised machine learning methods, particularly for low-dimensional ordinal IVs. While NNs are sophisticated and hold great potential for predicting outcomes with high-dimensional IVs, psychologists might also explore other advanced methods rooted in regression. Specifically, penalized linear regression algorithms, which balance bias and variance for improved performance, are noteworthy alternatives. Lasso (Tibshirani, 1996), Ridge (Hoerl & Kennard, 1970), and Elastic Net (Zou & Hastie, 2005) are three penalized linear regression models known to often surpass traditional linear regression in performance. While these models have fewer hyperparameters and coefficients, they might need a smaller sample size for a stable performance near the true value.

Finally, the absence of the bibliographical references section limits the ability to fully assess the academic rigour of the article.

In summary, the article ‘Estimating the Minimum Sample Size for Neural Network Model Fitting - A Monte Carlo Simulation Study’ addresses a relevant and pressing issue in the application of neural networks (NN) to psychometric data, offering valuable insights into the challenges and limitations associated with determining appropriate sample sizes. Although the study is methodologically robust and provides detailed discussions of its findings, there are areas where clarity, organisation and contextualisation could be improved. For example, the summary and introduction are more concise, technical terms and methodologies are better explained, and the discussion of practical implications and future lines of research is improved. In addition, the absence of a reference section considerably limits the assessment of the scientific basis of the article. If these aspects were addressed, I believe that the article could better serve the intended audience by providing clearer guidance and inspiring more specific research in this field.

Thank you very much for your suggestions for making this paper better!

Round 2

Reviewer 2 Report

Comments and Suggestions for Authors

Having reviewed the modifications made to the manuscript ‘Estimating the Minimum Sample Size for Neural Network Model Fitting with Psychometric Dataset - A Monte Carlo Simulation Study’, I believe that they have adequately addressed the observations raised during the review process.